# 3D Printing of Gelled and Cross-Linked Cellulose Solutions; an Exploration of Printing Parameters and Gel Behaviour

**DOI:** 10.3390/bioengineering7020030

**Published:** 2020-03-27

**Authors:** Tim Huber, Hossein Najaf Zadeh, Sean Feast, Thea Roughan, Conan Fee

**Affiliations:** 1School of Product Design, University of Canterbury, Private Bag 4800, Christchurch 8020, New Zealand; tlr50@uclive.ac.nz (T.R.); conan.fee@canterbury.ac.nz (C.F.); 2Biomolecular Interaction Centre, University of Canterbury, Private Bag 4800, Christchurch 8020, New Zealand; hossein.najafzadeh@pg.canterbury.ac.nz (H.N.Z.); sean.feast@pg.canterbury.ac.nz (S.F.); 3Department of Mechanical Engineering, University of Canterbury, Private Bag 4800, Christchurch 8020, New Zealand; 4Department of Chemical and Process Engineering, University of Canterbury, Private Bag 4800, Christchurch 8020, New Zealand

**Keywords:** bioprinting, cellulose, hydrogel, physical cross-linking

## Abstract

In recent years, 3D printing has enabled the fabrication of complex designs, with low-cost customization and an ever-increasing range of materials. Yet, these abilities have also created an enormous challenge in optimizing a large number of process parameters, especially in the 3D printing of swellable, non-toxic, biocompatible and biodegradable materials, so-called bio-ink materials. In this work, a cellulose gel, made out of aqueous solutions of cellulose, sodium hydroxide and urea, was used to demonstrate the formation of a shear thinning bio-ink material necessary for an extrusion-based 3D printing. After analysing the shear thinning behaviour of the cellulose gel by rheometry a Design of Experiments (DoE) was applied to optimize the 3D bioprinter settings for printing the cellulose gel. The optimum print settings were then used to print a human ear shape, without a need for support material. The results clearly indicate that the found settings allow the printing of more complex parts with high-fidelity. This confirms the capability of the applied method to 3D print a newly developed bio-ink material.

## 1. Introduction

3D printing, often also referred to as additive manufacturing, of biopolymers has attracted growing attention within the last few years. A large and increasing amount of 3D printing techniques is being developed, of which a large fraction fall within the categories of inkjet 3D printing, stereolithography printing and extrusion based 3D printing [1,2]. Currently, the most used polymers in 3D printing are polylactic acid (PLA), acrylonitrile butadiene styrene (ABS) and nylon, but a growing number of new materials have become available to be used on 3D printers, and both consumers and producers are developing printing materials based on natural polymers, driven by a need for more sustainable practices [3]. A comprehensive overview of 3D printing technology and available materials is beyond the scope of this work but is available for example in the work of [1] or [3].

Bioprinting as a subsection of 3D printing has also seen rapid and astonishing developments, and has been a major contributor towards the development of 3D printable natural materials and biopolymers, mostly in the form of hydrogels. In this case, the motivation for material development stems from a need for swellable, non-toxic, biocompatible and possible biodegradable materials also known as bio-ink materials, as the main application of bioprinting has been in tissue engineering and regenerative medicine [4,5,6,7]. A key aspect of bio-ink material development is the so-called biofabrication window, an attempt to describe the compromises that need to be made to achieve the highest print fidelity without compromising the ability of the printed material to function as a tissue scaffold [8,9]. This window is not only specific to the used bio-ink material but also depends on the used hardware including the used print setting, for example extrusion pressure, print speed or line height and width [10,11]. 

Polysaccharides such as alginate, hyaluronic acid or chitosan and proteins in the form of collagen and gelatine have been of particular interest and have been successfully printed into complex three-dimensional structures [12]. Most bioprinters rely on extrusion-based 3D printing, meaning the printing material needs to undergo a phase change from liquid to solid during the printing process. In bioprinting this often involves either a gelation process of a solution or, more commonly, thixotropic of a gel [13]. Cellulose is the most abundant biopolymer in the world and has been demonstrated to be a suitable polymer for applications in, for example, tissue engineering, filtration or separation [14]. However, cellulose in its native form is not water soluble and thus has proven challenging to process on bioprinters, either requiring the use of nanocellulose to form thixotropic and thus extrudable gels or suitable solvents to create cellulose solutions within a viscosity range that allows extrusion [15]. 

Only a few aqueous and non-aqueous solvents have been proven to be suitable for the processing of cellulose, but for most, the viscosity of the resulting solutions is not suitable for bioprinting or the solutions require a solvent exchange process to solidify the cellulose out of the solvent. Cellulose dissolved in the ionic liquid 1-ethyl-3-methylimidazolium acetate exhibited a shear thinning behaviour that made it suitable for extrusion printing [15] and a solution of cellulose in N-methylmorpholine-N-oxide (NMMO) has been successfully extruded on a bioplotter [16]. However, no other non-derivative cellulose solution has been reported to be printable by similar methods and thus little is known about the biofabrication window of cellulose. 

Cellulose can be readily dissolved in aqueous solutions of sodium hydroxide (NaOH) and urea, and while this solvent system is often described as attractive due its relatively low cost and low-toxicity, solubility is limited to no more than 6–8 wt.% depending on the type of cellulose used [17]. However, both, the solution viscosity and the properties of cellulose hydrogels made from this solvent can be adjusted by introducing additional amounts of undissolved cellulose particles as a physical cross-linker [18]. Cellulose solutions based on NaOH and urea also show a unique, irreversible gelation behaviour when heated [19,20]. This behaviour has been explored previously to 3D print non-derivatized cellulose using a focused laser beam to locally heat the cellulose solution beyond its gelation point [21]. While there is a significant amount of work published on the cellulose solution, little is known about the behaviour of the gelled solution [17]. In this work we will explore if the cellulose gel can be 3D printed using an extrusion based bioprinter to create three-dimensional objects and determine print related parameters of the biofabrication window. 

## 2. Materials and Methods

### 2.1. Chemicals

NaOH (99% purity) was purchased in pellets from The Sourcery (Christchurch, NZ). Urea (BioUltra, purity of 99.5%) and cellulose powder (Sigmacell (type 20, 20µm) were purchased from Sigma-Aldrich (St. Louis, MO, USA). All chemicals were used as received. 

### 2.2. Preparation of Cellulose Formulation

For each print session, deionized water (81 wt.%), urea (12 wt.%) and NaOH (7 wt.%) were measured and mixed using a Velp Scientifica (Usmate Velate, Italy) DLS overhead stirrer at 100 rpm [17,19,20,22]. After mixing for approximately 5 min, the mixture was heated to 50 °C under light stirring on a hot plate (Heidolph Instruments. Schwabach, Germany). After heating, 20 wt.% of cellulose was added to solvent mixture and stirred using a Velp Scientifica DLS overhead stirrer on 100–120 rpm for 2 minutes. The formulation was then cooled in a freezer set to −18 °C for 90 minutes to dissolve the cellulose. Subsequently, the solution was stored in a refrigerator at 5 °C until used for printing or rheometry measurements. This formulation was decided upon from previous trial and error experimentation, altering the weight percentage of the cellulose added to solution. 15, 20 and 25 wt.% cellulose solutions were prepared and extruded along a line from a 10 mL disposable syringe with 0.425 mm diameter and 0.5 mm length metal tip. The extruded gels were graded based on how well they held their shape and how easily they were extruded. From this basic test 20 wt.% cellulose was decided upon for further investigation.

### 2.3. Rheometry

Oscillatory measurements and rotational measurements were carried out using the Anton Paar Modular Compact Rheometer 302 (Anton Paar, Graz, Austria). For both measurements, the measuring system used was a parallel plate 50 (PP50) and the cell used was a Peltier plate. The temperature was set at 20 °C for all measurements. In total, 12 tests were carried out for the oscillatory measurements and 5 tests were carried out for the rotational tests. Each test was conducted with a cellulose formulation made no older than 5 days before being tested.

### 2.4. Oscillatory Measurements

For the oscillatory measurements, 4 different angular frequencies were tested 3 times each, the frequencies were 0.5 rad/s, 5 rad/s, 50 rad/s and 500 rad/s with a shear ramp of 0.1–1000%. For each test there were 41 data points plotted. Once the fresh sample was measured, the same sample was then sheared at constant shear strain of 100%, at 50rad/s for an interval of 120s. After the sample was sheared, the initial settings were used to measure the sheared sample. 

### 2.5. Rotational Measurements

1 test was conducted and repeated 5 times with a shear rate of 0.001–1000 1/s which plotted 91 data points with no set duration. The fresh sample was measured and then sheared using a constant shear rate of 100 1/s at a constant interval of 120 s. Once the sample was sheared, the settings from the initial measurements were used to measure the sheared sample. 

### 2.6. Printing Parameters

A Taguchi L9 orthogonal array [23] was used to reach the optimum 3D printing setting of cellulose gel with the minimum number of trials at a minimum cost. The optimum level of the process could be gained by applying a standard analysis. Since the aim of the experiments was to 3D print the nearly cubic shape of samples, a standard analysis was chosen to be “*nominal is the best*”. The experiment was conducted with multi-response results. 

Analysis of variance was done on the mean and Signal to Noise (S/N) ratio (Equation (1)). The collected measurements from the experiments were analysed using Minitab software (Minitab LLC, State College, PA, USA). The two important factors studied based on the preliminary tests were print pressure and print speed. The trialled settings are displayed in Table 1. The printing acceleration was kept constant to 4 [mm/s^2^] and printing line-width and line-height were set to 0.8 mm. The structure of Taguchi’s orthogonal robust design and the results of the measurement are shown in Table 2. The measured parameters were top surface area [mm^2^], side height [mm], side width [mm] and side view angles [°].
(1)$$S/N=-10\times \log s^{2}$$
where s is the standard deviation of the responses for all noise factors for the given factor level combination.

Cellulose formulations were prepared and loaded into a 16 mm diameter, 92 mm length barrel then centrifuged in an Eppendorf Centrifuge 5430 (Eppendorf, Hamburg, Germany) at 3000 rpm for 3 minutes to remove any air bubbles. After centrifugation, the barrel was then fitted with a 0.425 mm diameter and 0.5 mm length metal tip. The barrel was loaded into the Advanced Solutions Biobot (Life Sciences–Advanced Solutions, Louisville, KY, USA). For each trial, a cube with a side length of 10 mm was printed. All printing trials were carried out using Tissue Structure Information Modeling (TSIM^®^) software (Life Sciences– Advanced Solutions).

After a cube was printed, it was removed from the print stage and placed in 60 mL of deionized water for 24 hours to regenerate the cellulose. The water was changed regularly until full regeneration and solvent removal had occurred, as checked by measuring the pH of the water. After regeneration, the cubes were stored in a closed bottle of 60 mL deionized water and a spray of ethanol. 

After the 27 samples were completed, pictures were taken of the top view and side view of each cube using a Toupcam UCMOS01300KPA camera and ToupView software (Touptek, Zheijang, China) attached to a Nikon SMZ-1B confocal microscope (Nikon, Tokyo, Japan) for analysing. ImageJ Fiji software (National Institutes of Health, Bethesda, MD, USA) was used to analyse the cubes. The top surface area, width and length of the side view and all 4 angles of both top view and side view were measured and recorded. 

## 3. Results

### 3.1. Rheometry

To understand the behaviour of the created gels under shear, the viscosity of the gels was measured over a large range of shear stresses, to simulate stresses that occur during extrusion printing (Figure 1).

The gels show a continuous shear thinning behaviour, demonstrating a continuous alignment of the polymer chain in the gel. A behaviour that is also seen in ungelled solutions of cellulose [24]. Cellulose gels made from aqueous solutions of NaOH and urea are typically formed by the disruption of a so-called inclusion complex upon heating. In the solution state, the inclusion complex has been described as a sleevelike layer of solvent that surrounds the polymer chains and prevents interchain hydrogen-boding. When heated, those complexes appear to break down, allowing new hydrogen bonds to form resulting in rapid and irreversible gelation of the solution [25,26]. The added cellulose powder acts as a physical cross-linker promoting further hydrogen bonding and in turn stronger gels [18]. The shear thinning behaviour can be explained by the constant breakage and reformation of those hydrogen bonds between aligning polymer chains and the suspended cellulose particles. Similar behaviour was for example seen and deemed preferable for a bio-ink material based on carrageenan and suspended nanosilicates [27].

It is assumed that upon removal of the shear stress, a new gel will be formed and no permanent damage to the gel has occurred. This was further investigated by shearing the sample at a constant rate, followed by a shear sweep (Figure 1). While the initial viscosity of the presheared gel is lower than the original gel, likely due to chain alignment, the shear thinning behaviour is identical in the presheared gel, meaning no permanent damage to the gel has occurred. In turn, this means the viscosity of the gels can be easily modified to allow printing over a large range of shear stresses. 

To further investigate the effect of shear on the gel properties, oscillatory measurement over a range of shear strain were performed at different angular frequencies on unsheared and presheared samples (Figure 2). The sample showed a viscoelastic behaviour with the storage modulus G’ being larger than the loss modulus G” by approximately a factor of 10 at strains below approximately 10% for both tested gels as exemplarily shown for a frequency of 0.5 rad/s. At higher strains the gels start to yield leading to a cross over G’ and G” at higher strains. This yielding behaviour is common for colloidal gels and indicates initial gel rupture followed by a further break down of gel fragments into finely dispersed particles that allow the gel to flow [28]. An overall reduction between of G’ and G” is observable between unsheared and presheared gels, as well as a reduction of the loss factor (the ratio of G’: G”) by approximately 50% in the viscoelastic range. This supports the assumption that upon shearing, the gel is broken down into fine fragments, but upon removal of the strain, it is able to form a new, secondary gel through the extensively available hydrogen groups in the dissolved and suspended portions of cellulose, as well as interparticle forces between the dispersed cellulose particles and potentially gel fragments [29]. This in turn means that although the gel is likely to yield under the shear stresses during printing, the formation of a secondary gel will still allow a rapid solidification of the gel fragments into a secondary gel, and thus, allow for an extrusion of the used cellulose gels. 

There appears to be a small frequency dependence of the secondary gel regarding the cross-over point of G’ and G” (Figure 3) which can possibly be explained by a frequency dependence of the inter-particle interactions [28]. However, given the complexity of interaction in the tested systems, further studies that go beyond exploration of the printability of the used cellulose gels will be necessary to understand those interactions with more certainty. 

### 3.2. Taguchi Analysis of Printing Parameters

Not all used combination of printing factors yielded printed cubes suitable for an analysis of print fidelity. Overall, the variance of printing factors created prints with a large range of desirable and poor print fidelity (Figure 4).

The best 3D printed gel cube should have a flat surface with the closest surface area to 10 mm^2^, corners of 90°, side view width of 10 mm, side view height of 10 mm. By changing the print settings, the printed samples top area varied between 156.09 mm^2^ to 60.12 mm^2^ (Table 2).

The analysis of variance for S/N ratio (Table 3) and analysis of variance for means (Table 4) show that the pressure had a significant effect on 3D printing process with a contribution percentage of 99.9%, whereas speed has the minimum effect on the print quality with 25.7% contribution to the print quality. The R^2^ was calculated to 96.12%.

The F-ratio displayed in Table 2 represents the confidence in the collected data. Experimentation of the F-ratio in Table 3 and Table 4 show the control factor ‘pressure’ with an F-ratio value of 61.89 and 100.88, for S/N ratio and variance of means respectively, have a very low experimental error. 

The results from Taguchi analysis tests (mean and S/N ratio) are shown in Figure 5 and Figure 6. A higher S/N ratio, measured in decibels (dB), is preferred because a high value of S/N implies that the signal is much higher than the uncontrollable noise parameters, such as inconsistency in the data measurement.

The main effect graph is plotted by determining the means for each value of a categorical variable. The analysis of the mean value shows that pressure has a significant effect on 3D print quality. However, the response mean for speed is almost the same across all parameter levels (Table 5).

Based on the analysed results shown in Table 4, printing pressure has the highest impact on 3D printing a high-quality part. Speed has the lowest parameter impact on print quality. Thus, unlike trial-and-error experiments that implicitly rely on the experimenter’s judgment, Taguchi design of experiment has shown the best combination of controllable parameter levels. The best setting combination from the analysis of the results for 3D printing an object can be seen from the main effect of the plot for mean values (Figure 5) and the main effects plot for S/N ratio (Figure 6). These settings include pressure of 14 psi and speed of 5 mm/s. 

To validate the print settings found as optimal a complex three-dimensional shape in the form of an ear was printed including overhanging structures, typically difficult to achieve without the use of support material. The results (Figure 7) clearly indicate the found settings allow the printing of more complex parts with high-fidelity, and thus, confirm the success of the applied method to find optimal print settings for a newly developed bio-ink material. 

## 4. Conclusions

An investigation into the printability of a novel cellulose-based bio-gel formed from dissolving excess cellulose in a solution of urea and sodium hydroxide has been conducted. Rheological testing has demonstrated that the gel has shown two important properties needed by all extrusion-based inks. The gel is shear thinning and additional to this, forms of a secondary gel structure post shearing, allowing the gel to regain most of its structural integrity. The gel can therefore be easily extruded during printing, while post extrusion retaining its shape. The secondary gel formed may be weaker due to the initial yielding, alignment of polymer chains and dispersion of the gel fragments. This could impact the overall resolution of the printed gel due to slumping or reduced structural integrity of the gel. Further analysis into the gel’s behaviour is required to fully understand this complex system. Analyses of some of the many variables involved in bioprinting by the Taguchi DoE method allowed for optimization of the gel’s printability. Alteration of pressure was shown to have the greatest effect on the outcome of the print. This is not surprising as the extrusion rate is directly related to the pressure i.e., at higher pressure, more gel is extruded. Therefore, finer tuning of the pressure setting will be the most beneficial to improving the print quality. A more detailed study involving other variables, such as nozzle diameter and acceleration, may further improve printing accuracy and resolution. Optimal settings for pressure and speed were found and used to print a complex organic shape (human ear). The printing of this structure shows the cellulose gel was able to handle slight overhangs without support material, while maintaining shape fidelity. 

Prior to the development of this work cellulose was only able to be printed using solutions of ionic liquids or organic compounds [15,30]. This new bio-ink material offers a cheaper less toxic alternative to bioprinting using one of the most abundant materials found on earth. Overall, the printability of this new bio-ink material is incredibly promising as an inexpensive alternative to current cellulose based bio-ink materials. As with all new bio-ink materials, optimization of not only the formulation but also printer settings are required to ensure the best possible print. Here, only one formulation was tested; however, many variations are possible and will be tested in future works. 

## Figures and Tables

**Figure 1 bioengineering-07-00030-f001:**
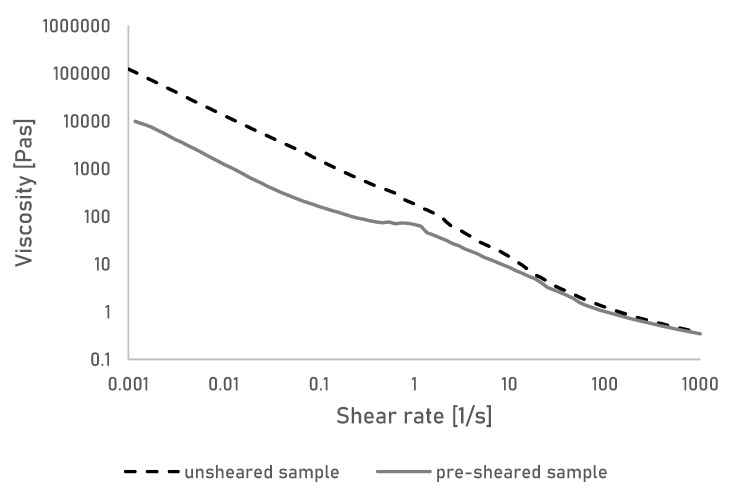
Viscosity of the cellulose gel measured over shear rate. Shown are results of a gel in unsheared and presheared state.

**Figure 2 bioengineering-07-00030-f002:**
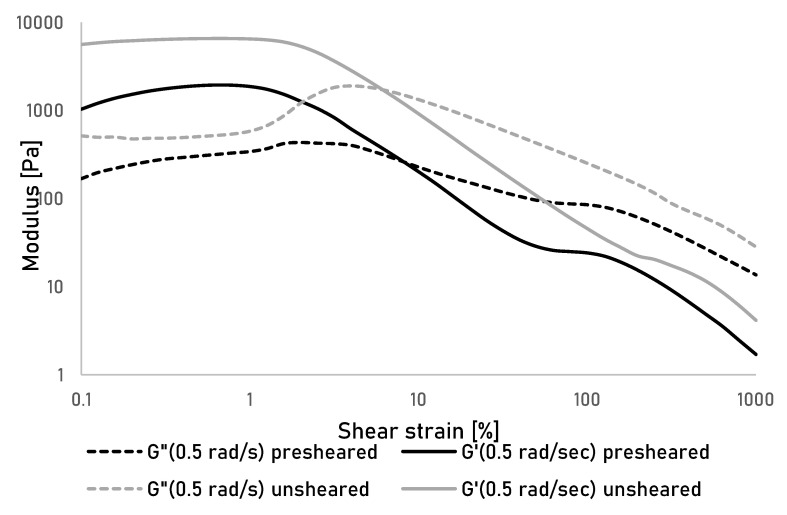
Storage (G’) and Loss (G") modulus of unsheared and presheared cellulose gels measured over shear strain at an angular frequency of 0.5 rad/sec.

**Figure 3 bioengineering-07-00030-f003:**
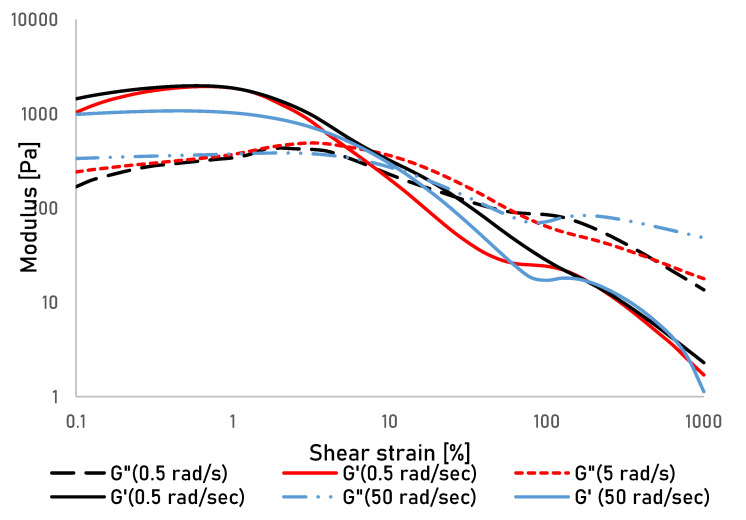
Storage (G’) and Loss (G") modulus of a presheared cellulose gel over shear strain measured at angular frequencies of 0.5, 5 and 50 rad/sec.

**Figure 4 bioengineering-07-00030-f004:**
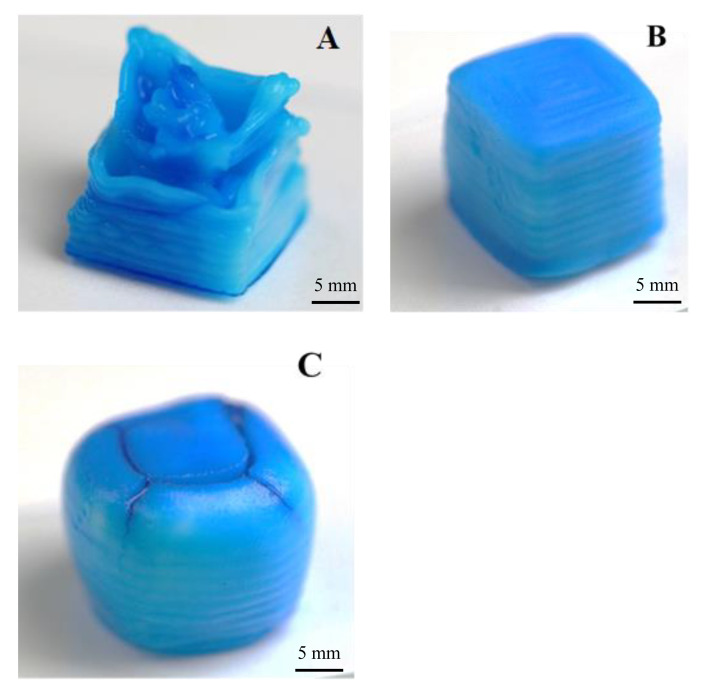
Examples of printed gel cubes with an approximate side length of 10 mm. Shown are a cube printed with insufficient extrusion of gel (**A**), overflowing amounts of extruded gel (**C**) and close to ideal print settings (**B**). Cubes have been dyed blue using food colouring for better contrast in the displayed photos.

**Figure 5 bioengineering-07-00030-f005:**
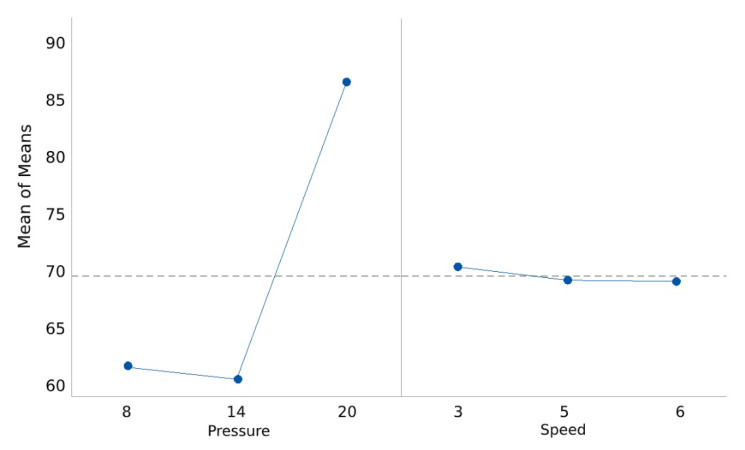
Main effect of the plot for mean values.

**Figure 6 bioengineering-07-00030-f006:**
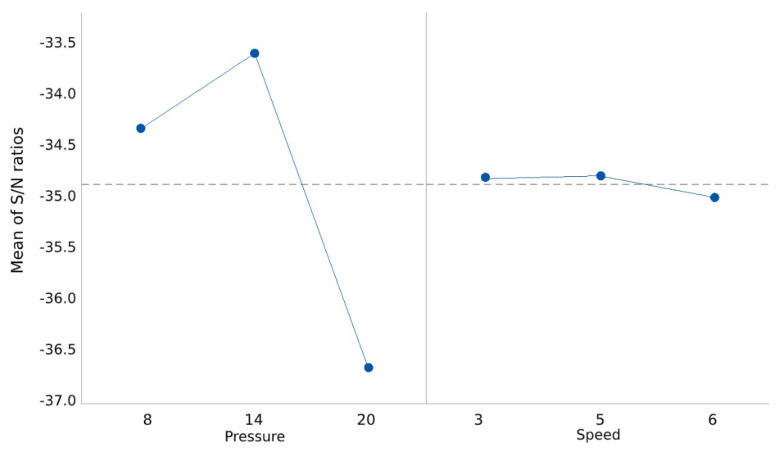
Main effects plot for S/N ratios.

**Figure 7 bioengineering-07-00030-f007:**
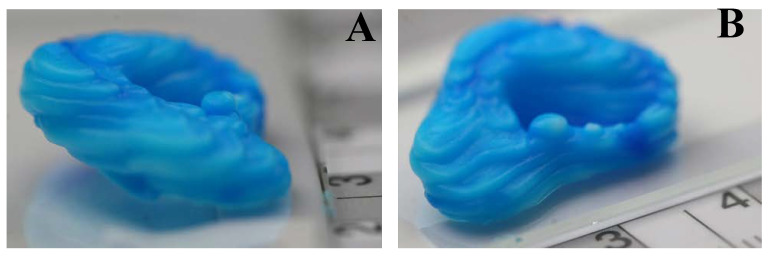
Test prints of a model of a human ear using optimal settings for extrusion pressure and print speed derived from the Taguchi analysis of the print parameters. (**A**) End view of test print; (**B**) Side view of test print.

**Table 1 bioengineering-07-00030-t001:** List of the conducted print trials at varying settings for extrusion pressure and print speed.

Trial	Pressure [psi]	Speed [mm/s]	Trial	Pressure [psi]	Speed [mm/s]
**1**	8	3	**10**	14	3
**2**	8	3	**11**	14	3
**3**	8	3	**12**	14	3
**4**	8	5	**13**	14	5
**5**	8	5	**14**	14	5
**6**	8	5	**15**	14	5
**7**	8	6	**16**	14	6
**8**	8	6	**17**	14	6
**9**	8	6	**18**	14	6
**19**	20	3	**23**	20	5
**20**	20	3	**24**	20	5
**21**	20	3	**25**	20	6
**22**	20	5	**26**	20	6
			**27**	20	6

**Table 2 bioengineering-07-00030-t002:** Results of the L9 orthogonal array tests and measurements.

Pressure [psi]	Speed [mm/s]	Top Area [mm^2^]	Side Height [mm]	Side Width [mm]	Angle 1 [°]	Angle 2 [°]
8	3	80.75	8.63	9.38	121.75	96.34
8	5	72.65	7.26	7.86	113.58	107.66
8	6	60.12	6.89	7.34	110.01	114.15
14	3	86.24	8.92	9.44	92.12	95.79
14	5	87.69	9.63	9.34	101.60	102.16
14	6	74.45	8.39	9.86	116.77	95.59
20	3	156.09	11.40	11.97	135.07	131.64
20	5	143.216	11.362	12.384	127.77	123.962
20	6	141.6	11.005	12.496	130.728	137.068
	**Max**	**156.09**	**11.4**	**12.49**	**135.07**	**137.06**
	**Min**	**60.12**	**6.89**	**7.34**	**92.12**	**95.59**

**Table 3 bioengineering-07-00030-t003:** Analysis of variance for SN ratios.

Source	DOF	Sum of Square	Variance	F-ratio	P
**Pressure**	2	16.2566	16.2566	61.89	0.001
**Speed**	2	0.0839	0.0839	0.32	0.743
**Residual Error**	4	0.5253	0.5253		
**Total**	8	16.8659			

**Table 4 bioengineering-07-00030-t004:** Analysis of variance for means.

Source	DOF	Sum of Square	Variance	F-ratio	P
**Pressure**	2	1295.73	1295.73	100.88	0.000
**Speed**	2	2.98	2.98	0.23	0.803
**Residual Error**	4	25.69	25.69		
**Total**	8	1324.39			

**Table 5 bioengineering-07-00030-t005:** Response table for S/N ratio.

Level	Pressure	Speed
**1**	−34.33	−34.82
**2**	−33.57	−34.80
**3**	−36.72	−35.01
**Delta**	3.15	0.21
**Rank**	1	2

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
