# Peer review of "3D Printing of Gelled and Cross-Linked Cellulose Solutions; an Exploration of Printing Parameters and Gel Behaviour"

_bioengineering, 2020, doi:10.3390/bioengineering7020030_

Round 1

Reviewer 1 Report

The manuscript presented by Tim Huber et.al. presents an interesting approach to 3D print the most abundant natural polymer of cellulose in its soluble form as the feeding ink. The main findings have provided adequate scientific insights and technical details for the society of bioink printing where natural polymers are applied. The manuscript is well-written and the discussion is presented in depth. It can be published as such, in my opinion. 

Author Response

We thank the reviewer very kindly for the review. The document has been spell-checked and very minor corrections have been carried out where necessary.

Reviewer 2 Report

In this work, the authors have used cellulose gels for 3D printing and its optimization using Taguchi method. Also, rheological assessments of the gels were carried out. The work looks good and may be accepted for the publication in this journal. However, I have a few questions and comments for the authors.

Comments:

  1. Line 94: Freezing to dissolve,? Please clarify. If this is not the actual process, kindly modify the procedure section.
  2. 2. methods: Please cite the references for the ratio (de-ionized water (81 wt.%), urea (12 wt.%) and NaOH (7 wt.%)) used for the gel preparation. As it was reported already by others. (e.g. 2006, Cai et al etc???)
  3. Line 104. “they were” words are hanging in between? Modular Compact Rheometer 302 (Anton Paar, Graz, Austria),” they were.”
  4. Line 94 “The formulation was then put in a freezer set to -18°C for 90 minutes to dissolve the”
  5. Table 1: The headings are too close and very hard to read (suggestion foot note: and mark the headings with codes). Also, merge the numbers inside the table to make it easy for the readers.
  6. Figure 3. difficult to distinguish among the lines within the figure. Kindly consider revising the figures with different designs to distinguish the lines.
  7. Figure 4 and 6. Please include the scale bars.
  8. Figure 6. Design to printed structure difference should be calculated and presented (dimensions and deviations from the actual design). This will give a better idea about the optimized gel printing conditions and its fidelity.
  9. I strongly suggest the authors to express the gels as “biomaterial ink” and not “bioink” in the manuscript as no cells are used in the study. Please see the recent publications for the definitions. Consider, revising the title too, as it is misleading. “Bioplotting”
  10. I also suggest the authors to cite more papers from the recently published articles from 2019 and 2020. As recent works are more vital for the discussion section.
  11. Please check the punctuations throughout the manuscript. E.g. 301, 299, 79-line number.

Author Response

Line 94: Freezing to dissolve,? Please clarify. If this is not the actual process, kindly modify the procedure section.

This has been clarified.

 methods: Please cite the references for the ratio (de-ionized water (81 wt.%), urea (12 wt.%) and NaOH (7 wt.%)) used for the gel preparation. As it was reported already by others. (e.g. 2006, Cai et al etc???)

Additional references have been added.

Line 104. “they were” words are hanging in between? Modular Compact Rheometer 302 (Anton Paar, Graz, Austria),” they were.”

Line 94 “The formulation was then put in a freezer set to -18°C for 90 minutes to dissolve the”

Please check the punctuations throughout the manuscript. E.g. 301, 299, 79-line number.

This has been corrected together with other minor spelling mistakes.

Table 1: The headings are too close and very hard to read (suggestion foot note: and mark the headings with codes). Also, merge the numbers inside the table to make it easy for the readers.

The table has been updated to improve readability.

Figure 3. difficult to distinguish among the lines within the figure. Kindly consider revising the figures with different designs to distinguish the lines

The figure was updated.

Figure 4 and 6. Please include the scale bars.

A scale bar was added to figure 4. We don't see the necessity to ad a scale bar to figure 6 as it is a plot to show the signal to noise ratio of the analysis. Assuming the reviewer meant Figure 7 we would like to draw the reviewer's attention to the included rules in the photo which serves as a scale bar.

Figure 6. Design to printed structure difference should be calculated and presented (dimensions and deviations from the actual design). This will give a better idea about the optimized gel printing conditions and its fidelity.

See above, assuming the reviewer meant Figure 7 we do not see the additional benefit provided through this as it would only be an approximation anyway.Figure 7 shows the result of a final test print with the previously determined best printing condition to show that those setting allow the printing of complex shapes including overhangs.   

I strongly suggest the authors to express the gels as “biomaterial ink” and not “bioink” in the manuscript as no cells are used in the study. Please see the recent publications for the definitions. Consider, revising the title too, as it is misleading. “Bioplotting”

This comment was strongly appreciated by the authors as we are determined to ashere and use concurrent definitions. The expression has been changed accordingly throughout the manuscript.

I also suggest the authors to cite more papers from the recently published articles from 2019 and 2020. As recent works are more vital for the discussion section.

Additional references were added.